# Resistance Trends in *Klebsiella pneumoniae* Strains Isolated from Bloodstream Infections in a Tertiary Care Hospital over a Period of 7 Years

**DOI:** 10.3390/microorganisms13112451

**Published:** 2025-10-25

**Authors:** Alina Maria Borcan, Elena Rotaru, Georgiana Radu, Elena Liliana Costea, Calin Andrei Borcan, Mihaela-Cristina Olariu, Madalina Simoiu

**Affiliations:** 1Faculty of Medicine, The University of Medicine and Pharmacy “Carol Davila”, Dionisie Lupu Street, No. 37, 050474 Bucharest, Romania; alina.borcan@umfcd.ro (A.M.B.); mihaela.olariu@umfcd.ro (M.-C.O.); madalina.simoiu@umfcd.ro (M.S.); 2The National Institute of Infectious Diseases “Prof. Dr. Matei Balș”, Doctor Grozovici Street, No. 1, 021105 Bucharest, Romania; 3Fundeni Clinical Institute, Soseaua Fundeni, No. 163, 022328 Bucharest, Romania

**Keywords:** *Klebsiella pneumoniae* bloodstream infections, blood cultures, multi-drug-resistant *Klebsiella pneumoniae*, carbapenem-resistant *Klebsiella pneumoniae* (CRKP)

## Abstract

*Klebsiella pneumoniae* is one of the top pathogens causing bloodstream infections (BSIs) worldwide. The rise of carbapenem-resistant *K. pneumoniae* (CRKP) and multidrug-resistant (MDR) strains is of particular concern as therapeutic options are limited. Analyzing local resistance profiles is essential for the success of antibiotic stewardship strategies. This study aims to explore the resistance profiles of *K. pneumoniae* strains identified in BSI in a tertiary care hospital over 7 years. Automated systems were used to test antibiotic susceptibility. Results were interpreted according to EUCAST clinical breakpoints. The rate of multidrug resistance (MDR) was 57.6%. The percentage of ESBL producers was 54.5%, and the percentage of carbapenemase producers was 43.2%. Overall resistance rates to other antibiotics were 47.1% to ciprofloxacin, 31.4% to gentamicin, 25.7% to amikacin, 20.9% to colistin, 19.6% to Fosfomycin, and 44.5% to trimethoprim/sulfamethoxazole. The highest resistance to colistin was recorded in 2023 (28%). More than half of the strains in the study were MDR and ESBL producers. *K. pneumoniae* resistance to colistin has increased during the last 7 years. The rates of carbapenemase-producing bacteria (CPB) are on the rise. The most frequently co-harboring carbapenemases were NDM and OXA-48. Local antibiotic resistance rates are crucial in implementing an effective antibiotic stewardship strategy.

## 1. Introduction

Invasive infections represent prevalent conditions within healthcare environments, with sepsis and septic shock continuing to pose significant health challenges worldwide [1]. Data published in 2020 showed that there were 48.9 million cases and 11 million sepsis-related deaths globally, representing 20% of all global deaths [2]. Moreover, for every 1000 hospitalized patients, an estimated 15 patients will develop sepsis as a complication of receiving healthcare [2]. Bacteremia is defined as the presence of viable bacteria in the bloodstream as proven by positive blood culture results, and will lead to sepsis in most cases if left untreated. Invasive infections are potentially fatal complications in all patients, but children, the elderly, pregnant, and immunocompromised patients are at a higher risk. Any infectious agent can lead to invasive infections, but the most common causes are bacterial [3,4]. The most common bacteria incriminated in bloodstream infections (BSI) are Staphylococcus aureus, Escherichia coli, *Klebsiella pneumoniae*, Pseudomonas aeruginosa, Acinetobacter baumannii, and Streptococcus pyogenes [5]. Due to impressive advances in medicine in the past decades and patients having longer life spans due to more efficient treatments, there has also been a rise in the number of immunocompromised individuals and invasive procedures contributing to the high incidence of BSI. As the only treatment for bacterial BSI is antibiotic agents, losing their power in the face of growing resistance rates will only lead to a higher burden on healthcare systems and patients all around the world. Developing newer antimicrobial agents is a lengthy and expensive process, which leads to the development of costly molecules that are prohibitive to poorer countries, at least during the first years. There is consensus in the medical and scientific community that nosocomial infections caused by multidrug-resistant bacteria constitute a silent pandemic. According to the World Health Organization (WHO), they could cause close to 10 million deaths every year by 2050 [6].

*Klebsiella pneumoniae* is a Gram-negative, encapsulated, rod-shaped, non-motile bacterium from the Enterobacteriaceae family. It represents an important opportunistic pathogen that causes a large number of nosocomial infections, particularly in developing countries. Urinary tract infections (UTIs), bloodstream infections (BSIs), respiratory tract infections (RTIs), and surgical site infections are frequent severe infections that can be caused by *K. pneumoniae* [7]. Ten years ago, it was estimated that *K. pneumoniae* caused over 2 million serious infections in healthcare settings, including 351,000 cases of BSI [6]. A rise in antimicrobial resistance to antibiotics, particularly carbapenems, has complicated the treatment of infections caused by *K. pneumoniae*. Prior to 2001, almost 99.9% of clinical Enterobacteriaceae strains were susceptible to carbapenems. The latest Global Antimicrobial Resistance and Use Surveillance System (GLASS 2022) report by the WHO states that in 2020, very high levels of resistance in pathogens causing BSI were reported, regardless of testing coverage. The 2024 WHO Bacterial Priority Pathogens List (WHO BPPL) includes third-generation cephalosporin-resistant Enterobacterales (3GCRE) and carbapenem-resistant Enterobacterales (CRE) in the critical priority category. This emphasizes their burden and need for targeted interventions, especially in low and middle-income countries [8,9,10].

The expanded β-lactamase resistance (ESBL) in *Klebsiella pneumoniae* involves the overexpression of efflux pumps, decreased outer membrane permeability due to changes in porins, and the production of enzymes such as β-lactamases, which inactivate antibiotics. Target site mutations and biofilm formation also enhance resistance by protecting against antimicrobials and immune responses [11].

The majority of CPE strains are resistant to commonly used antibiotics. Hence, treating infections caused by these microorganisms often requires the use of new antimicrobials or combinations of two, sometimes three, antibiotics. Currently, there are more than 8000 β-lactamases described in the medical literature, classified in 1980 by Ambler based on their functional features and amino acid sequences. Thus, classes A, C, and D use a serine as an enzyme active center, whereas β-lactamases of class B use the metal zinc (also known as metallo-β-lactamases). Carbapenemases are β-lactamases that are encoded by genes on chromosomes or plasmids and fall within one of three Ambler classes (A, B, D). Current therapeutic options for CPKP (and other CRE) vary according to the Ambler class of carbapenemases. Thus, for class A or D carbapenemases (particularly KPC and OXA-48 enzymes), tigecycline, colistin, ceftazidime/avibactam, or cefiderocol are viable options. In contrast, for class B carbapenemases (particularly NDM, one of the most potent carbapenemases in terms of carbapenem hydrolysis), the possibilities are ceftazidime/avibactam with aztreonam or cefiderocol. Novel BL/BLI combinations have been implemented successfully worldwide: ceftazidime/avibactam, meropenem/vaborbactam, imipenem/relebactam, imipenem/cilastatin/relebactam, and aztreonam/avibactam. Unfortunately, the emergence of strains co-harboring carbapenemases from different classes (for example, NDM and KPC) results in higher levels of antibacterial resistance, further limiting treatment options. Moreover, the emergence of newer, more potent variants of carbapenemases, such as the new metallo-beta-lactamases NDM-1 and NDM-5, also makes treatment challenging. Additional resistance mechanisms, such as porin mutations and active efflux pumps, may accompany carbenemases and decrease susceptibility to novel antibiotics and combinations. Local interventions such as directing empiric antibiotic prescribing should be linked to local resistance prevalence, but in practice, this is not always the case [12,13,14].

*Klebsiella pneumoniae* is also a primary species capable of acquiring AmpC-type cephalosporinase, also known as class C Ambler β-lactamase, which is typically located on plasmids. These enzymes hydrolyze penicillins, cephalosporins (specifically third-generation, not fourth-generation compounds), and monobactams. The classical ESBL inhibitors do not inhibit these enzymes effectively [15].

RCP bacteria are isolates that carry two or more carbapenemases. They represent a new and highly dangerous threat, as few treatment options we have for CPE may become obsolete in the presence of hydrolysis enzymes from different classes. These new strains of harmful bacteria are widely distributed worldwide, and their number is increasing over time. There are more than 100 strains currently reported globally, from countries such as China, South Korea, Turkey, Iran, and India. The most common strains of RCP are *K. pneumoniae* and Acinetobacter baumannii. They go so far as to exhibit varying proportions of carbapenemase combinations, such as the new endemic strain of *K. pneumoniae* harboring NDM and OXA-48 enzymes. Another recently reported combination is between KPC and VIM (Verona integron-encoded metallo-beta-lactamase). Moreover, the highest mortality from RCP strains is in respiratory system infections, followed by BSI [16]. In our study, the prevalence of RCP *K. pneumoniae* has grown to a point where in the last year of the study, more than half of the CPKP isolates for which we could identify the enzyme harbored more than one carbapenemase. The most frequent co-harboring of carbapenemases in the present study was between NDM and OXA-48.

*K. pneumoniae* can harbor and express carbapenemases—carbapenemase-producing *Klebsiella pneumoniae* (CPKP), the most potent beta-lactamases, which are capable of hydrolyzing newer carbapenem drugs used in the treatment of MDR bacterial infections. *K. pneumoniae* carbapenemase (KPC) is one of the most common carbapenemases, causing outbreaks all over the world. They are the most prevalent carbapenemases in the Unites States of America [17]. Other common carbapenemases harbored by *K. pneumoniae* causing outbreaks at this point are New Delhi Metallo-beta-lactamases (NDM) and OXA-48. In Europe, national outbreaks of CPKP have been reported mainly in Southern European countries such as Greece, Spain, and Italy, where the highest prevalence is also observed [18]. Co-harboring of two or more carbapenems in the same strain have also been identified, particularly NDM and OXA-48. These particular isolates, known as redundant carbapenemase-producing (RCP) bacteria, are a growing threat worldwide. BSI caused by carbapenem-resistant Enterobacterales (CREs) have higher mortality rates than infections caused by carbapenem-susceptible strains [19].

Antibiotic stewardship is a global strategy aimed at improving habits of prescription and use of antibiotics, in order to increase the effectiveness of available antimicrobial treatments, protect patients from harms caused by unnecessary antibiotic use, and combat antimicrobial resistance (AMR). There is no “one size fits all” approach to optimizing antibiotic use for all settings worldwide. Several sectors (human, animal, and agricultural) inappropriately or unnecessarily use antibiotics, making the solution to this issue complex and demanding. Several local factors influence the effectiveness of antimicrobial stewardship strategies, including antibiotic prescribing habits and awareness of antibiotic resistance rates, which vary significantly even at the national level. One of the recommended strategies to effectively counteract AMR is conducting surveillance campaigns aimed at improving the selection of appropriate empirical antibiotic treatment locally.

This study analyzed the antibiotic resistance and beta-lactamases’ production rates for *K. pneumoniae* strains identified in blood cultures from patients hospitalized in the National Institute of Infectious Diseases “Prof. Dr. Matei Balș”, Bucharest, Romania, from the 1 January 2017, to the 31 December 2023, for a period of 7 years.

## 2. Methods

### 2.1. Sample Collection and Processing

The most common method for determining the etiology of bacteremia is through the use of blood cultures, as they are highly sensitive and easy to use in clinical settings. The study analyzed all *K. pneumoniae* strains identified in blood cultures from patients of all ages, who were hospitalized in the National Institute of Infectious Diseases “Prof. Dr. Matei Balș”, Bucharest. All blood culture sets (consisting of one aerobic and one anaerobic bottle) were collected according to the hospital’s standard operating procedures, with the volume of blood in each vial ranging between 8 and 10 mL. Samples were incubated in the BACT/ALERT 3D (BioMérieux, Salt Lake City, UT, USA) automated system for seven days. Bottles flagged as positive were removed from the equipment, Gram-stained, subcultured on solid media, and further incubated in both aerobic and anaerobic conditions at a temperature of 35 ± 2 °C. Microbiology laboratory personnel notified the primary team prescriber via phone regarding the Gram-stain result of any positive blood culture within 30 min of a positive outcome. Plates were examined according to laboratory procedures every 24 h. Bacteria were identified using the Vitek 2 Compact (BioMerieux, Salt Lake City, UT, USA) and the automated matrix-assisted laser desorption/ionization (MALDI) TOF MS (Bruker, Billerica, MA, USA) systems.

### 2.2. Antibiotic Susceptibility Testing

Antimicrobial susceptibility testing (AST) was carried out in accordance with the established operating procedure. The methods used for AST were broth microdilution, performed with automated systems such as VITEK 2 (bioMérieux) and MICRONAUT (Bruker Daltonics GmbH & Co., KG, Bremen, Germany). The following antibiotics were tested: amoxicillin-clavulanate (AMC), piperacillin-tazobactam (TZP), aztreonam (ATM), imipenem (IPM), meropenem (MER), ertapenem (ETP), ceftriaxone (CRO), ceftazidime (CTZ), cefepime (CFPM), ciprofloxacin (CIP), gentamicin (GM), amikacin (AMK), colistin (COL), and trimethoprim-sulfamethoxazole (TMP-SMX). AST was performed using the Vitek2 Compact (BioMerieux, Salt Lake City, UT, USA) and MICRONAUT-AM (Bruker, Billerica, MA, USA) systems. Suspicion of beta-lactam antibiotic resistance enzyme production was raised according to EUCAST guidelines for detecting resistance mechanisms and specific resistances of clinical and/or epidemiological importance. ROSCO Diagnostica (Taastrup, Denmark) kits were used for definitive confirmation of all bacteria. NG-Biotech Laboratories’ immunochromatography tests (Guipry, France) were also used for the identification of ESBL (only CTX-M) and carbapenemase-producing strains. Quality control for COL was performed in accordance with EUCAST guidelines. Results were interpreted annually in accordance with the updated EUCAST clinical breakpoints. Multidrug resistance was defined as resistance to at least one agent in three or more categories of antimicrobials. To facilitate analysis, the microorganisms classified as susceptible “S” and susceptible, increased exposure “I” in this study were grouped together. Data were organized and interpreted using Microsoft (MS) Excel (version 2019, Redmond, WA, USA).

## 3. Results

A total of 36.134 blood cultures were processed in the study period. The total rate of positivity was 7.1% (n = 2.573). Of these, 8.8% (n = 228) were identified as *Klebsiella pneumoniae*. It was the third most common pathogen identified in blood cultures in our laboratory, after *Escherichia coli* and *Staphylococcus aureus*, and the second most common Gram-negative bacterium. The highest incidence was in the year 2017 (n = 44), while the lowest number of strains was recorded in 2021 (n = 16). In 2020 and 2022, *K. pneumoniae* represented more than 10% of all pathogens identified in positive blood cultures (Figure 1). More than half of patients with *K. pneumoniae* BSI were male (62.7%). The average age of patients in the study was 66.81 [14–101], with a median of 70. The mortality rate was 25% (n = 57). The highest mortality was recorded in the year 2021 (n = 10).

Overall resistance to combinations of beta-lactams and beta-lactamase inhibitors (BL/BLI) was as follows: 54.8% to amoxicillin/clavulanic acid and 50% to piperacil-lin/tazobactam. The resistance rates were on a general upward trend, peaking in 2021. However, the rates of resistance in 2023 are still higher than those registered before the COVID pandemic (Figure 2).

Overall resistance to cephalosporins during the study period was of 54.3% to ceftazidime, 53.0% to cefotaxime and 42.5% to cefepime. The year 2021 marks the peak of resistance to third- and fourth-generation cephalosporins, 100% of isolates being resistant to ceftazidime and the resistance rates to fourth-generation cephalosporins surpassing 50% for the first and only time during our study (Figure 3).

Overall resistance to monobactams was of 32.0% for aztreonam, while to carbapenems was of 34.2% to imipenem, 36.8% to ertapenem and 42.1% to meropenem (Figure 4).

Overall resistance to other antibiotics was as follows: 47.3% to ciprofloxacin, 31.5% to gentamicin, 25.8% to amikacin, 21.0% to colistin, 19.7% to fosfomycin and 44.3% to tri-methoprim/sulfamethoxazole. Resistance to colistin has varied between 15 and 25% during the first 6 years of the study, and has reached 28% for the first time in 2023. Resistance rates to trimethoprim/sulfamethoxazole has increased continuously with a peak in 2022, followed by a decrease in 2023 (Figure 5).

In what concerns resistance to beta-lactam antibiotics through secretion of beta-lactamases, the overall prevalence of extended-spectrum beta-lactamases (ESBL) producers is of 55.2% and resistance to carbapenems (CRKP) is 44.7%. Multidrug-resistance (MDR) was identified for 57.6% of strains included in the study. The highest rates of resistance were recorded in the year 2021, with 100% of strains being CRKP and MDR. On the other hand, the lowest rate of MDR was recorded in 2017, of 39%.

The type of carbapenemase was identified in 45% (n = 46) of carbapenem-resistant *K. pneumoniae* (CRKP) strains. Of these, 65.2% (n = 30) were expressing NDM, 41.3% (n = 19) were secreting OXA-48 and 26.0% (n = 12) were secreting KPC. 28.2% (n = 13) were secreting more than one carbapenemase, and 44.3% (n = 101) of all strains were secreting both ESBL and carbapenemases. Concerning the RCP strains, 21.7% are harboring NDM and OXA-48 simultaneously, 2.1% are harboring NDM and KPC, and 4.3% are harboring NDM, OXA-48, and KPC. In 2023, 66.6% (n = 10) of strains for which carbapenemases were identified showed co-harboration of two or even three enzymes simultaneously. The most frequent combination was the production of NDM and OXA-48, observed in 80% (n = 8) of the strains. From the total of CPKP strains identified in 2022 and 2023, enzymes were detected in 86% (n = 19) and 93.7% (n = 15) of these, respectively, as NDM, OXA-49, or KPC.

## 4. Discussion

The EUROBACT-2 international cohort study found that Gram-negative bacteria are predominant pathogens in BSI in the ICU worldwide, with *K. pneumoniae* being the most frequent. The same results were obtained by Meera et al. in a study from India [20,21]. Similar results were obtained by colleagues from China, India, Colombia, Taiwan, as well as the GLASS 2022 study, *K. pneumoniae* being the second most frequent Gram-negative rod involved in BSI, after *E. coli.* [9,19,22,23,24]. In our laboratory, Gram-negative rods are also the most prevalent agents, with E. coli being the most frequent, followed by *K. pneumoniae*. The male/female ratio in the current study was 1.66. Similar results are mentioned in available literature, with ratios varying from 1.41 to 1.64 [22,23,25,26,27]. One study from Saudi Arabia had a higher incidence in female patients [28].

There is a high heterogeneity in the prevalence of antibiotic resistance for *K. pneumoniae* isolates from around the world, not just in the case of BSI. Comparing different resistance rates, as well as reporting global pooled rates for antibiotic resistance, is of great aid in being aware of recent trends and preparing for future challenges in the context of the global circulation of resistance genes.

Rates of resistance to BL/BLI vary between 60% to amoxicillin/clavulanic acid in Saudi Arabia and Brazil, 27% to piperacillin/tazobactam in Colombia and 37.5% in Brazil, and 78.8% to ticarcillin/clavulanic acid in India [23,28,29]. In our study, we can clearly see a dramatic increase in resistance to traditional BL/BLI combinations, more to amoxicillin/clavulanic acid than piperacillin/tazobactam, which peaked in 2021. However, in 2023, more than half of the strains were still resistant to these antibiotic combinations. Resistance to third- and fourth-generation cephalosporins exceeds 50% in many areas around the world, particularly in the context of a growing ESBL-harboring rate that has been on an upward trend over the past two decades. Overall, third-generation cephalosporin resistance in *K. pneumoniae* strains from bloodstream infections varies from 18% in Colombia, to 25% in Greece, 36% in Saudi Arabia, 73.5% in India, and 40–54% in China. The GLASS 2022 found high levels of resistance to third generation cephalosporins, varying between 59 and 64%, and fourth-generation cephalosporins, of 57.4%, reported in *K. pneumoniae* strains worldwide [21,23,28,30,31]. In our study, more than half of strains were resistant to third generation cephalosporins and more than one third were resistant to fourth-generation cephalosporins, less than the global prevalence. However, this result is caused by a very high resistance rate to third-generation cephalosporins in 2021, when there were only 16 strains isolated in blood cultures from patients hospitalized during the COVID-19 pandemic, most probably of nosocomial nature. Analyzing the yearly rates of resistance to cephalosporins, less than half of the isolates are resistant to cephalosporins each year.

According to the ECDC Antimicrobial Resistance Surveillance in Europe 2023 report, which presents data from 2021, resistance to third-generation cephalosporins in *Klebsiella pneumoniae* has become increasingly widespread across the WHO European Region. In 2021, only seven of the forty-five reporting countries (16%) exhibited resistance rates below 10%. In contrast, nineteen countries (42%), predominantly situated in the southern and eastern parts of the region, reported resistance levels of 50% or higher. Carbapenem resistance was more frequently detected in *K. pneumoniae* than in *Escherichia coli*. In 2021, resistance levels in the northern and western areas of the WHO European Region remained comparatively low, with fourteen of forty-five countries (31%) reporting rates below 1%. Conversely, fifteen countries (33%) reported resistance levels of 25% or higher, of which eight exhibited rates equal to or exceeding 50% [32].

Pooled global resistance in *K. pneumoniae* strains from bloodstream infections to carbapenems is 44.4% to ertapenem, 35.2% to imipenem, and 36.1% to meropenem [33]. Resistance to gentamicin varies from 17% in Colombia to 29.6% in Saudi Arabia, and to amikacin from 7% in Colombia to 18.4% in Saudi Arabia [23,28]. Resistance to gentamicin for the isolates in the present study was higher than rates available in the literature. In contrast, rates of resistance to amikacin are comparable. The global pooled resistance rate to aminoglycosides is 33.3% for gentamicin and 25.4% for amikacin [33]. Resistance to ciprofloxacin seems to be quite similar around the world, with little difference, ranging from 34% in Colombia to 36.8% in Saudi Arabia [23,28]. The overall global pool resistance was recently reported as 45.3% [33]. In our study, the rate of resistance to fluoroquinolones was higher, with almost half of the strains showing resistance to ciprofloxacin.

Current therapeutic options for MDR Gram-negative pathogens, particularly in *K. pneumoniae*, *E. coli*, Pseudomonas aeruginosa, and Acinetobacter baumannii, are so extremely limited that clinicians are forced to use older, previously discarded drugs, such as colistin, as a last resort. Unfortunately, they are associated with significant toxicity. Moreover, resistance to colistin has started to emerge and grow worldwide, with the appearance of the MCR-1 gene in China in 2015 [34], which is a plasmid-encoded gene and highly mobile. As it is a mobile genetic element, it is transmitted horizontally to other strains and even species of bacteria. The pooled global resistance to colistin in strains from BSI was reported recently by colleagues Uzairue et al. as 3.1%. Resistance to colistin varies from 6.6% in Saudi Arabia to 19.2% in Thailand, to 16.6% in East China to 0.8% in South Korea. There is a significant increase in global colistin resistance in isolates collected starting in 2020. The same study found a resistance rate of 2.8% reported in isolates before the year 2015, of 2.9% in isolates reported between 2016 and 2019 and almost a 5× increase up to 12.90% after the year 2020. Moreover, colistin resistance rates are higher in bloodstream infections from intensive care unit patients (11.5% versus 3.0%) [28,33,35]. These results are consistent with our findings, with a rate of colistin resistance on the rise in the past years. Other recent rates of resistance to trimethoprim/sulfamethoxazole reported in available literature are of 52.6% in Saudi Arabia and 37% in Colombia. The GLASS report showed global pooled rates of resistance of 34.3–61.3% [9,23,28]. In our study, resistance to trimethoprim/sulfamethoxazole was close to 45%. Surveilling resistance rates for trimethoprim/sulfamethoxazole and colistin is important not only in *K. pneumoniae* isolates, but also in other Enterobacterales and Gram-negative rods, as genes of resistance for these important antibiotics are stored on plasmids, being mobile genetic elements, thus increasing the likelihood of spreading horizontally and contributing to generating pan-drug-resistant and extreme-drug-resistant nosocomial strains.

The growing rate at which antibiotics were used to treat infections in the past years has led to an increase in prevalence of MDR bacteria, particularly *Klebsiella pneumoniae*. A recent meta-analysis by colleagues Asri et al. analyzed the global prevalence of nosocomial MDR *K. pneumoniae*, showing that the pooled global prevalence is estimated at 32.8%. However, the results were heterogenous, ranging from 98.1% in South Korea, to 72.4% in South America, 31.2% in Europe to 12.9% in North America, 66.4% in Shanghai, China and the lowest prevalence of 3.1% in Tunisia. Another recent study reported a pooled global MDR prevalence of *K. pneumoniae* isolates from BSI of 80.1% [7,33,35]. In our study, the results are conclusive with the global trend, the overall rate of MDR has surpassed half of the strains analyzed.

Carbapenems were intended to be used as reserve antibiotics for treating infections that are difficult to treat. However, due to an increase in patients with carbapenemase-producing organisms, the use of carbapenems has increased globally. Global CRKP varies with region, from 89.6% in China, 35.2% in Brazil, 24.6% in the United States of America, 3.2% in Canada, and 2.1% in Mexico, 12.2% in Argentina, and lower in Europe and Australia: 1.2% in the United Kingdom, 0.14% in Northern Ireland, 1.4–3.7% in Australia. The highest prevalence rates of CRKP in Europe are in Greece, Bulgaria, and Romania, three bordering countries [10,36]. This is consistent with results from our study, where more than 40% of strains were CRKP, and the rate of MDR surpassed half of the strains. Isolates resistant to carbapenems are usually MDR and have high rates of treatment failure [9]. In Europe, most CRKP are antibiotic resistant due to carbapenemase production mechanisms. This is why we consider active surveillance of resistance enzymes a crucial step in combating their spread and implementing a successful antibiotic stewardship strategy tailored to local patterns, as genes are encoded by mobile genetic elements and are easier to eliminate in the absence of constant antibiotic pressure [10]. The most prevalent carbapenemase among CPKP worldwide is KPC [37]. For KPC, the global prevalence ranges from 17.0% to 33.3% [7,38]. For the isolates in our study for which we could identify the carbapenemase, more than one-fourth of them harbored KPC.

In our hospital, an antibiotic stewardship strategy for BSI was implemented in March 2023. For Enterobacterales, the first line of treatment consists of ampicillin (only for *E. coli*), ampicillin/sulbactam, amoxicillin/clavulanic acid, ceftriaxone, cefuroxime, gentamicin and trimethoprim/sulfamethoxazole. The second line consists of ertapenem, ceftazidime, amikacin, piperacillin/tazobactam, and aztreonam, while the third line consists of meropenem, imipenem, colistin, tigecycline, ceftazidime/avibactam, ceftolozane/tazobactam, intravenous fosfomycin, and cefoderocol. Further studies comparing resistance rates before and after the implementation of the antibiotic stewardship strategy are encouraged. The study is limited by its single-center nature. Moreover, patients treated in our hospital primarily come from the southern and eastern regions of Romania; thus, the results of our study reflect the antibiotic resistance trends in this region.

During the pandemic period, the National Institute of Infectious Diseases “Matei Bals” served as a referral point for COVID-19 cases only. The majority of ICU-admitted patients had comorbidities and accounted for most of the complicated cases within the period. The ward patients were frequently admitted through transfers from hospitals around the country due to the inability to provide adequate treatment.

The post-pandemic period marked the beginning of the admission of multiple infectious diseases, as well as the containment of raw strains of *Klebsiella pneumoniae.*

## 5. Conclusions

More than half of the strains in the study were MDR and ESBL producers. *K. pneumoniae* resistance to colistin is increasing. The rates of carbapenemase-producing bacteria (CPB) are on the rise. The most frequent co-harbored carbapenemases in our study were NDM and OXA-48. Local antibiotic resistance analysis is essential in implementing an effective antibiotic stewardship strategy.

## Figures and Tables

**Figure 1 microorganisms-13-02451-f001:**
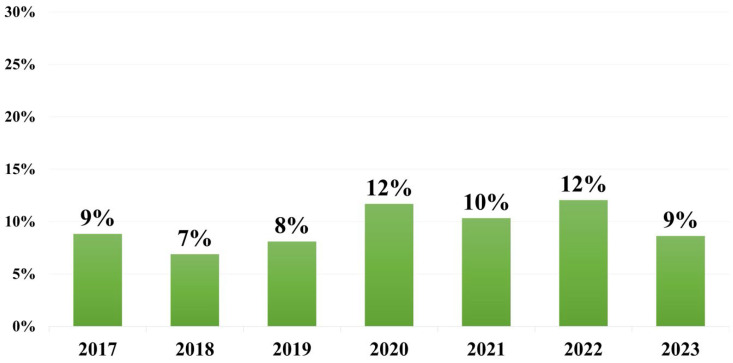
Percentage of *K. pneumoniae* strains identified in positive blood cultures each year.

**Figure 2 microorganisms-13-02451-f002:**
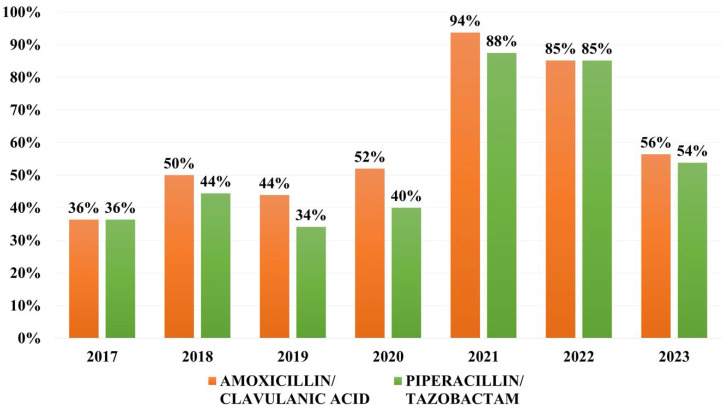
Distribution of resistance rates to combinations of beta-lactams and beta-lactamase inhibitors (BL/BLI).

**Figure 3 microorganisms-13-02451-f003:**
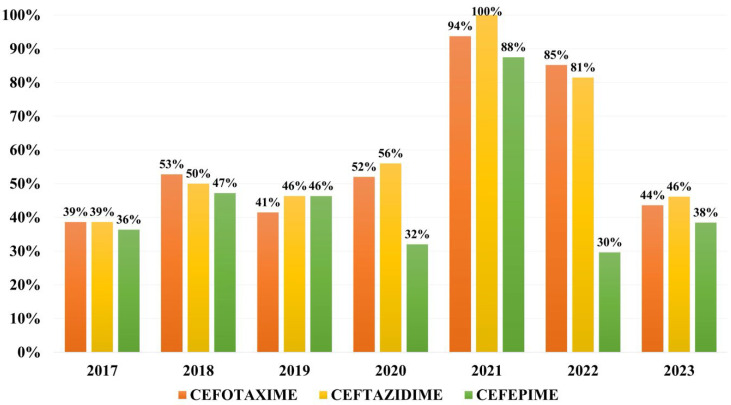
Distribution of resistance rates to 3rd and 4th generation cephalosporins.

**Figure 4 microorganisms-13-02451-f004:**
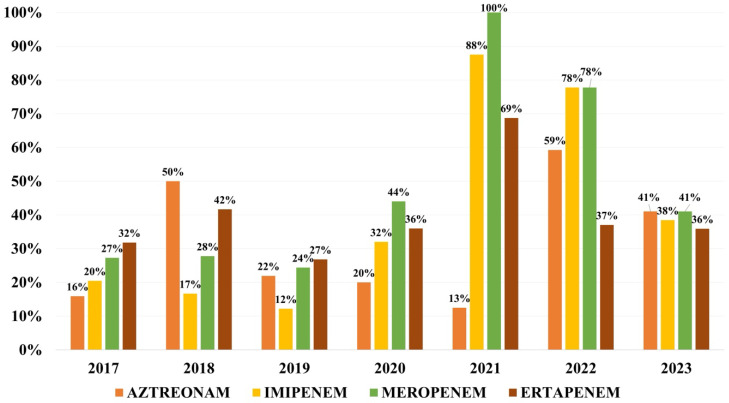
Distribution of resistance rates to monobactams and carbapenems.

**Figure 5 microorganisms-13-02451-f005:**
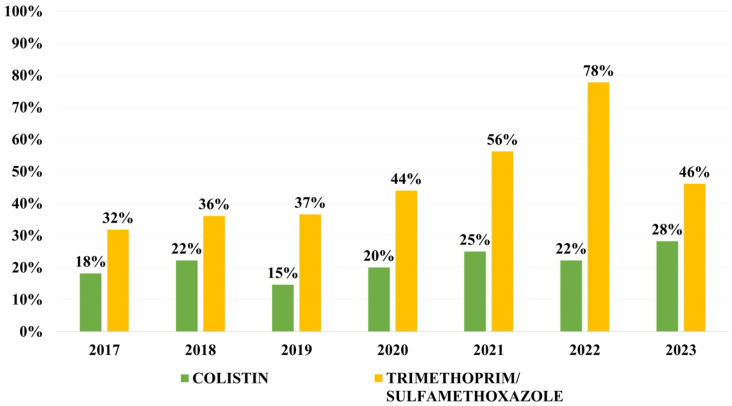
Distribution of resistance rates to trimethoprim/sulfamethoxazole and colistin.

## Data Availability

The original contributions presented in this study are included in the article. Further inquiries can be directed to the corresponding author.

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
