# Peer review of "Resistance Trends in Klebsiella pneumoniae Strains Isolated from Bloodstream Infections in a Tertiary Care Hospital over a Period of 7 Years"

_microorganisms, 2025, doi:10.3390/microorganisms13112451_

Round 1

Reviewer 1 Report

Comments and Suggestions for Authors

An extremely important and significant topic from a clinical perspective. A significant follow-up period and the range of currently applicable antibiotics to which Klebsiella spp. are susceptible are included, so it is important to praise the idea and the research itself!

- Add some answers:

1. What is the prevalence of acute kidney injury and complicated urinary tract infections during the specified period?

2. Were aminoglycoside antibiotics administered?

3. Was the antibiotic dose determined and by what
method?

4. How did the inflammation parameters change?

- WBC, CRP, Pct and fibrinogen!

Author Response

Dear Reviewer,

We would like to sincerely thank you for your thoughtful and constructive feedback on our manuscript titled “Resistance trends in Klebsiella pneumoniae strains isolated from bloodstream infections in a tertiary care hospital over a period of 7 years”. We appreciate the opportunity to communicate and gain a comprehensive understanding of our research results.Your comments are immensely helpful in enhancing the quality and clarity of our work. Below is a response outlining the changes made and the reasons for any suggestions that were not incorporated.

1.An extremely important and significant topic from a clinical perspective. A significant follow-up period and the range of currently applicable antibiotics to which Klebsiella spp. are susceptible are included, so it is important to praise the idea and the research itself!

2.. What is the prevalence of acute kidney injury and complicated urinary tract infections during the specified period?

  1. Were aminoglycoside antibiotics administered?
  2. Was the antibiotic dose determined and by what
    method?
  3. How did the inflammation parameters change?

- WBC, CRP, Pct and fibrinogen!

Answer: Thank you in advance for your valuable insights. We agree that such a clinical perspective would provide a better and broader picture; however, the data available for this particular retrospective study were limited to the microbiological laboratory, without additional patient information. We would be happy to incorporate your suggestions for future research in collaboration with our Infectious Diseases clinicians, who could offer a more comprehensive clinical dataset.

Reviewer 2 Report

Comments and Suggestions for Authors

Resistance trends in Klebsiella pneumoniae strains isolated from bloodstream infections in a tertiary care hospital over a period of 7 years

The study aimed to analyze resistance trends in Klebsiella pneumoniae isolates from bloodstream infections in Romania. Over 50% of the isolates were mulidrug-resistant. There was 54% of ESBL producers and 43% carbapenemase producers.

The introduction section does not provide a comprehensive understanding of the cephalosporin and carbapenem resistance mechanisms. More informations are necessary in the introduction section to better understand the magnitude of carbapenem and  and colistin resistance and therapeutic impact. Moreover, the topic was researched extensively before and there are a lot of references covering this issue.  The methods are insufficiently described. The conclusions do not indicate clearly if the purpose of the study has been achieved.  The conclusions contain general statements about the need for infection control and antibiotic stewarship. Instead of that the authors should comment on the relevance of their findings.  The manuscript should be better presented in laboratory techniques, data presentation, writing and discussion.

MAJOR COMMENTS

  1. More informations about extended-spectrum beta-lactamases, plasmid-mediated AmpC beta-lactamases and carbapenemases should be provided in the introduction section. Other resistance mechanisms were not mentioned such as porin loss or hyperexpression of efflux pumps.
  2. In the material and methods section the authors should explain and describe the method for ESBL detection. Was it double disk-synergy test, combined disk test with clavulanic acid, dilution method or E test. It is not clear.
  3. Statistical analysis should be done to determine if there is statistically relevant difference in the resistance rates between years.
  4. In the results section, there is too much listing of the data with little point.
  5. There is no explanation in the discussion section for the decline of ESC and carbapenem resistance in the postpandemic period.

MINOR COMMENTS

  1. Title: Resistance trends in Klebsiella pneumoniae strains isolated from bloodstream infections in a tertiary care hospital over a 7 Years period
  2. The genus and species names of bacteria should be italicised throughout the text
  3. Page 2, line 48: Staphylococcus instead of Staphy-lococcus
  4. Page 1, line 27 trimethoptim instead trime-thoprim
  5. Page 9, lines 316-335: the discussion section should be focused on the results achieved in the study. In this section the authors explain to the reader what the carbapenemases are and how they are classified. This should be moved to the introduction section. The same pertains to the therapeutic options.

Comments on the Quality of English Language

The quality of English language should be improved. The paper should be proofread by a native English speaker

Author Response

Dear Reviewer,

      We would like to sincerely thank you for your thoughtful and constructive feedback on our manuscript titled “Resistance trends in Klebsiella pneumoniae strains isolated from bloodstream infections in a tertiary care hospital over a period of 7 years” Your comments are immensely helpful in enhancing the quality and clarity of our work. We have carefully reviewed each suggestion and will revise the manuscript accordingly. Below is a detailed, point-by-point response outlining the changes made and the reasons for any suggestions that were not incorporated.

  1. More information about extended-spectrum beta-lactamases, plasmid-mediated AmpC beta-lactamases,and carbapenemases should be provided in the introduction section. Other resistance mechanisms were not mentioned, such as porin loss or hyperexpression of efflux pumps.

Answers: Thank you for this helpful suggestion. We agree and will provide the necessary modifications to the introductory section, clarifying the mechanisms of resistance relevant to ESBL strains. In the material and methods section, the authors should explain and describe the method for ESBL detection.

  1. Was it double disk-synergy test, combined disk test with clavulanic acid, dilution method or E test. It is not clear.

Answer: The methods used for AST were broth microdilution with automated systems such as VITEK® 2 (bioMerieux) and MICRONAUT® (Bruker Daltonics GmbH & Co. KG), wich provided with information on ESBL expextation status. And were confirmed of ESBL presence by immunochromatography tests NG-TEST® CTX-M Multi (NGBiotech, France) or supplemented by ROSCO Diagnostica (Taastrup, Denmark) kits using the disk diffusion method.

  1. Statistical analysis should be done to determine if there is a statistically significant difference in the resistance rates between years.

Answer:  The data collected from the microbiological laboratory and was analyzed by clinicians without involving a professional statistician.

  1. In the results section, there is an excessive listing of data with little significance.

Answer: We aimed to present the most relevant findings using the accessible information on the strains isolated in our laboratory, where a clear trend of changes in resistance rates was observed.

  1. There is no explanation in the discussion section for the decline of ESC and carbapenem resistance in the post-pandemic period.

Answer: We will include further explanation. During the pandemic period, the National Institute of Infectious Diseases “Matei Bals” served as a referral point for COVID-19 cases only. The majority of ICU-admitted patients had comorbidities and accounted for most of the complicated cases within the period. The ward patients were frequently admitted through transfers from hospitals around the country due to the inability to provide adequate treatment.

The post-pandemic period marked the beginning of the admission of multiple infectious diseases, as well as the containment of raw strains of Klebsiella pneumoniae.

Thank you again for your time, expertise, and valuable insights.

Kind regards,

Elena Rotaru MD

Medical Microbiologist

on behalf of all authors.

Round 2

Reviewer 1 Report

Comments and Suggestions for Authors

No

Author Response

Dear reviewer,

We express our gratitude for your valuable suggestion. In response, we have incorporated additional details regarding the AmpC cephalosporinase characteristics of the Klebsiella pneumoniae strains. The amended text now includes information on AmpC production, which can be found in the introduction section.

Yours sincerely,

On behalf of all the authors,

Dr. Elena Rotaru

Medical Microbiologist.

Reviewer 2 Report

Comments and Suggestions for Authors

The authors have mentioned only ESBLs as the resistance mechanisms to expanded-spectrum cephalosporins, but failed to  mention plasmid-mediated AmpC beta-lactamases which hydrolyze third generation cephalosporins and cephamycins and spare fourth generation cephalosporins.

Author Response

(The authors gave the same response as above.)
